# Investigation on the influence of unbalanced shaft component in gearbox on displacement using the Newmark-β method

Thanh Lam Tran[1], Vinh Phoi Nguyen [2], Chi Cuong Le[1], Thien Ngon Dang[1]*

**1** Faculty of Mechanical Engineering, HCMC University of Technology and Education, Ho Chi Minh, Viet Nam, **2** Faculty of Engineering and Technology, Pham Van Dong University, Quang Ngai, Viet Nam

* ngondt@hcmute.edu.vn

## Abstract

This study introduces an enhanced numerical approach for analyzing the dynamic behavior of a rotor-bearing system subjected to unbalanced excitation from a gearbox drive shaft. The Newmark-β method with the integration of a variable time-step algorithm was used, allowing the system to be solved rapidly and accurately without compromising stability. This technique enables a precise computation of displacement and torsional deformation of the rotating shaft during its operational cycle. The proposed computational model is validated against experimental data, showing deviations of displacement in normal operation below the critical speed of about 6%. A comprehensive parametric analysis is conducted to evaluate the influence of rotational speed, trial mass, and initial phase angle on the system dynamics. The findings confirm that our enhanced numerical approach yields rapid convergence and reliable predictions, making it a valuable tool for dynamic analysis of rotating systems.

## 1. Introduction

The drive shafts in gearboxes are common components with variety of applications in industries, operating at various speeds and load types. Among many factors causing failure of shafts, vibration due to unbalance is a major cause. Single and coupled vibration modes, including torsional, longitudinal and transverse vibrations induce fatigue, fracture and tribological issues on the rotating shaft components [1–3]. These vibrations result in displacement, performance of gear transmission, wear and cracks [4–6].

Many studies have been carried out on dynamic aspects of the of rotor systems. The modelling of the rotor systems often uses the Jeffcott rotor model having a massless axis and a mass disk placed in the middle of the shaft. Modern analytical methods have been utilized to provide a foundational understanding of rotor dynamics and to conduct simple model experiments [7,8]. In addition, [9] and [10] studied a rotor system with a dynamic model, in which a disc is placed in the middle of a massless

**Data availability statement:** All relevant data are within the manuscript and its Supporting information files.

**Funding:** This work belongs to the project in 2025 funded by Ho Chi Minh City University of Technology and Education, Vietnam.

**Competing interests:** The authors have declared that no competing interests exist.

elastic rotating shaft. The equations of motion are obtained to Lagrangian dynamics for transverse – torsional vibrations. [11] derived the equation of motion by assuming that the diesel engine drive system can be approached as a simple rotor model such as the Jeffcott rotor. A modified version of this rotor model was also used for analysing the coupled torsional–transverse vibrations of a propeller shaft resulting from misalignment induced by shaft rotation [12]. Besides, [13] modelled a flexible rotating shaft system subject to bending and torsion coupled with the shaft and disk moving away from the center point of the shaft. In these studies, it is evident that shaft displacement caused by vibration is a complex problem. Therefore, it is necessary to investigate the shaft's displacement in detail due to unbalance, using a new approach and model in the study of rotor dynamics.

The Newmark-β method, that is a widely used numerical integration technique in finite element analysis, is particularly effective for simulating dynamic systems. A study by [14] used this method to analyze the dynamic characteristics of spur and helical gear systems, while a subsequent study focused on the influence of housing flexibility on gear transmission dynamics [15]. An improved Newmark-β method was applied in [16] to determine nonlinear dynamics and reduce crankshaft torsional vibration. An enhanced version of the method was proposed for long-term simulations [17], offering improved convergence in evaluating the effects of nonlinearities on engine crankshaft torsional behavior. Collectively, these studies confirm that the Newmark-β method is a suitable and reliable tool for addressing time-varying oscillation problems.

In general, the current studies lack experimental evidence to support the theoretical computation. This study represents a computational dynamic model for a rotor-bearing system under unbalanced excitation, using the Newmark-β method with a variable time-step algorithm to determine the unbalance amount and shaft displacement. Various key experimental parameters, including rotational speed, trial mass, and initial phase angle, are put into account. The displacement and the amount of unbalance, determined from experiment, are compared with the numerically computational values to verify the validity of the computational model and thus to evaluate their effects on system dynamics. The motional orbit, representing shaft displacement, is also determined to evaluate the influence of the unbalance on the deflection of the shaft. This will allow to predict the fatigue strength, thus to accurately predict performance during operation.

## 2. Dynamic model of the rotor system

### 2.1 The proposed model

This issue arises from the eccentricity and imbalance of rotating components. If the gearbox operates under long-term conditions, it may lead to wear and eventual failure due to fatigue fracture. Fig 1 shows a shaft assembly in the gearbox of a lathing machine, comprising the drive shaft 3, bearings 1 and 5, and gears 2 and 4. Fig 2 represents the schematic diagram of a generalized model comprising a shaft and various assembled disks.

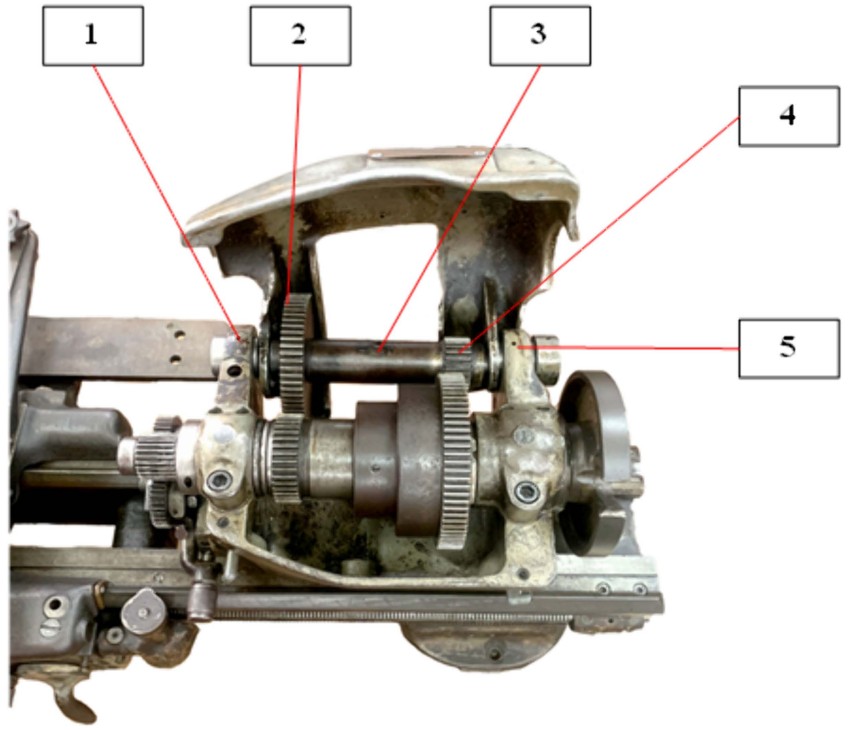

**Fig 1. Gear shaft assembly in the lathe gearbox.**

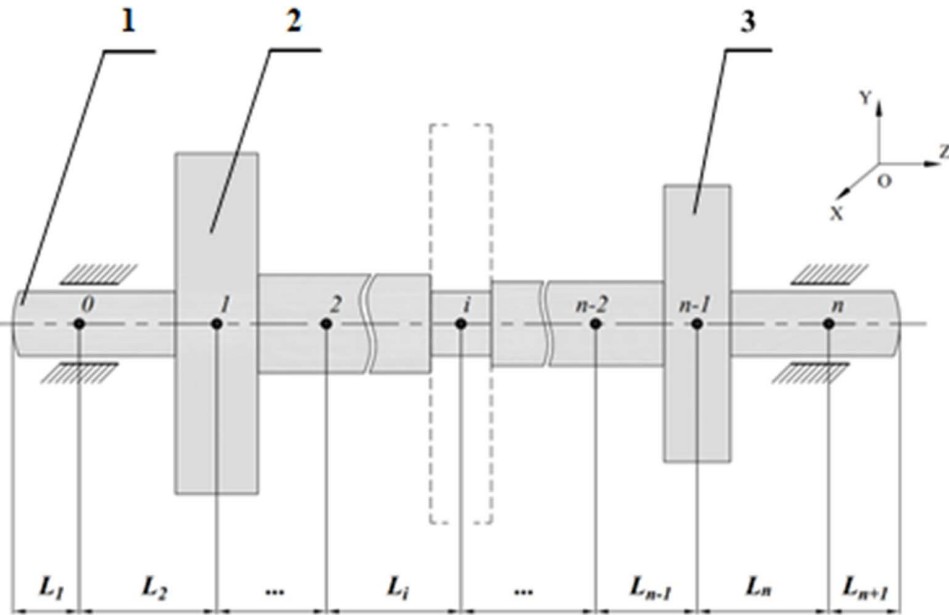

*1. Rotating shaft – 2. Disk 1 – 3. Other disks*

**Fig 2. Schematic diagram of the rotor–bearing system.**

## 2.2 Parameters of the model

The proposed model of transmission shaft with two disks is shown in Fig 3. The specimens for evaluating the fatigue strength of metallic materials were used, according to ISO 1143:2010. This specimen type is also well-suited for evaluating location to failure [18]. Seven nodes, from node 1 to node 7, on the rotating shaft are selected to be investigated. Nodes 1 and 7 are the bearing positions. Nodes 2 and 6 are the disk placements and nodes 3 and 5 correspond to the positions with the maximum cross-section on the shaft. Node 4 has the smallest cross-section, where the fracture occurs.

The displacement vector $q_i$ for the shaft nodes i (i = 1–7) is:

$$q_i = [x_i, y_i, \alpha_i]^T \tag{1}$$

where the displacements $x_i$ and $y_i$ along the X- and Y-axes and torsional angle $\alpha_i$ at the investigated positions are

$$x_i = [x_1, x_2, x_3, x_4, x_5, x_6, x_7]^T; \ y_i = [y_1, y_2, y_3, y_4, y_5, y_6, y_7]^T$$

$$\alpha_i = [\alpha_1, \alpha_2, \alpha_3, \alpha_4, \alpha_5, \alpha_6, \alpha_7]^T$$

$$\{x_i = x_n \ or \ x_i = x_m + e_m \cos \varphi_m \ and \ y_i = y_n \ or \ y_i = y_m + e_m \cos \varphi_m \ :$$

$$n = 1, \ 3, \ 5, \ 7; \ m = 2, \ 4, \ 6$$

Since the loads are applied onto the gears at nodes 2 and 6, the rotation angle on the shaft segment between nodes 2 and 6 are thoroughly investigated. The torsional angles at nodes 3 and 5 are assumed to be small and negligible $(\alpha_3 = \alpha_5 = 0)$. Thus, the torsional angles at nodes 2, 4, and 6 are $\alpha_i = [\alpha_2, \alpha_4, \alpha_6]^T$ where i = 2, 4, 6. The governing equation for the rotation angle is given as:

$$\varnothing_i(t) = \alpha_i + \omega_i t + \varnothing_{0_i} \tag{2}$$

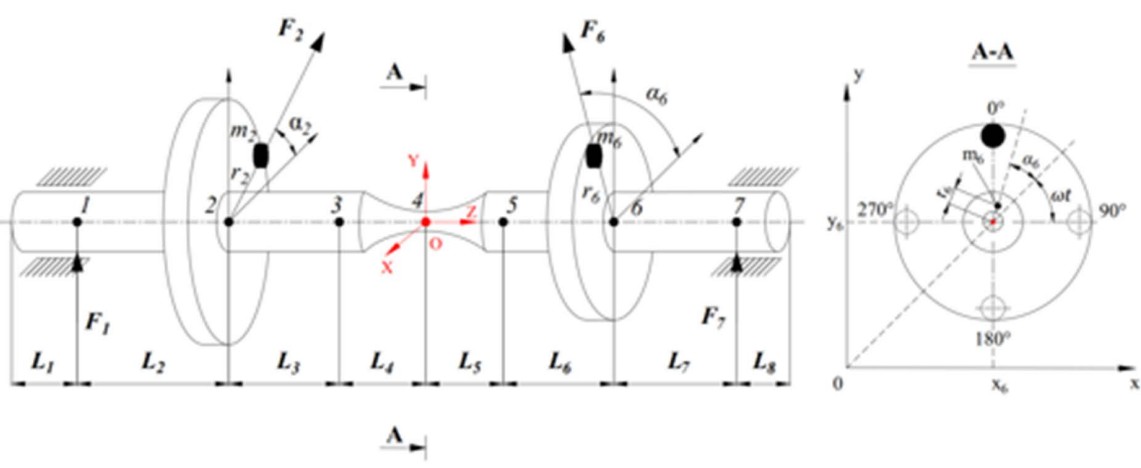

a) Shaft element model with two disks

b) Cross-section showing eccentric mass location

**Fig 3. Forces and unbalance positions of rotor-bearing system with eccentric masses.**

where $\varnothing_i(t)$ is the total rotation angle at node $i$ as time $t$; $\varnothing_{0_i}$ is the initial phase angle at node $i$; $\alpha_i$ is the torsional angle at node $i$; $\omega_i$ is the angular velocity at node $i$ ($i = 2, 4, 6$). The kinetic energy of the system is represented as:

$$T = T_t + T_r = \frac{1}{2}\{\dot{x}\}^T[M_c]\{\dot{x}\} + \frac{1}{2}\{\dot{y}\}^T[M_c]\{\dot{y}\} + \frac{1}{2}\{\dot{\varphi}\}^T[J_c]\{\dot{\varnothing}\} \tag{3}$$

where $T_t$ is translational kinetic energy; $T_r$ is rotational kinetic energy; $\{\dot{x}\}$, $\{\dot{y}\}$ are the vectors of translational velocities in the X and Y directions; $\{\dot{\varnothing}\}$ is the vector of angular velocities of the components; $[M_c]$ is the mass matrix corresponding to the system from node 1 to node 7; $[J_c]$ is the static moments of inertia concerning the study nodes 2, 4, 6. The elastic potential energy of the system is:

$$V_c = \frac{1}{2}\{x\}^T[K_x]\{x\} + \frac{1}{2}\{y\}^T[K_y]\{y\} + \frac{1}{2}\{\alpha\}^T[K_t]\{\alpha\} \tag{4}$$

where $[K_x]$ and $[K_y]$ are respectively the stiffness matrices corresponding to translational stiffness in the X- and Y-axes, $[K_t]$ is the torsional stiffness matrix related to angular displacements. According to Fig 3, an examination of the shaft segment from node 2 to node 6 represented by $I_3$ to $I_6$ was performed. Fig 4 shows the spring modeling of the shaft segment.

In the torsional shaft problem, $x_2$, $x_4$, and $x_6$ are equated to the torsional angles $\alpha_2$, $\alpha_4$ and $\alpha_6$ of the shaft at nodes 2, 4 and 6, respectively. From the modelling of the shaft segment in Fig 4, the elastic potential energy is written as:

$$V = \frac{1}{2}k_{t_2}\alpha_2^2 + \frac{1}{2}k_{t_4}(\alpha_2 - \alpha_4)^2 + \frac{1}{2}k_{t_6}(\alpha_4 - \alpha_6)^2 \tag{5}$$

The dissipated energy of the system is given by:

$$D = \frac{1}{2}[\dot{x}]^T[C_x]\{\dot{x}\} + \frac{1}{2}[\dot{y}]^T[C_y]\{\dot{y}\} + \frac{1}{2}[\dot{\alpha}]^T[C_t]\{\dot{\alpha}\} \tag{6}$$

where $D$ is the total energy dissipated due to damping in the system; $[C_x]$, $[C_y]$, and $[C_t]$ are respectively the damping matrices corresponding to translational damping in the X, Y directions and axial torsional damping. The dissipated energy due to damping is written as:

$$D = \frac{1}{2}C_{t_2}\dot{\alpha}_2^2 + \frac{1}{2}C_{t_4}(\dot{\alpha}_2 - \dot{\alpha}_4)^2 + \frac{1}{2}C_{t_6}(\dot{\alpha}_4 - \dot{\alpha}_6)^2 \tag{7}$$

where $C_{t_2}$, $C_{t_4}$ and $C_{t_6}$ are the torsional damping at nodes 2, 4, and 6. The Lagrange's equations of the second kind can be written in terms of the system as follows [19]:

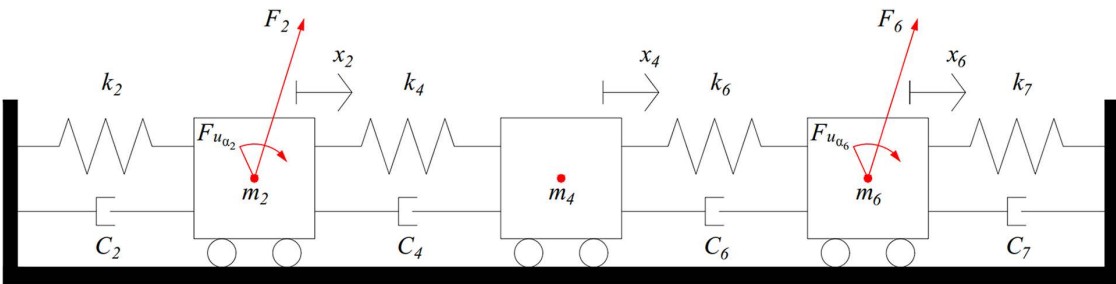

**Fig 4. Modelling of the study shaft segment from nodes 2, 4 and 6.**

$$L = T - V \tag{8}$$

$$\frac{d}{dt}\left(\frac{\partial T}{\partial \{\dot{q}_i\}}\right) - \frac{\partial T}{\partial \{\dot{q}_i\}} + \frac{\partial D}{\partial \{\dot{q}_i\}} + \frac{\partial V}{\partial \{\dot{q}_i\}} = F_i$$

where T is the total kinetic energy, V is the total potential energy of the system; $F_i = F_{g_i} + F_{u_i}$; $F_{g_i}$ is force due to the mass of the node; $F_{u_i}$ is centrifugal force due to unbalance. The differential equation of motion for the rotor at this time is:

$$[M]\{\ddot{q}\} + [C]\{\dot{q}\} + [K]\{q\} = \{F_i\} = \{F_{g_i}\} + \{F_{u_i}\} \tag{9}$$

where $[M]_{17x17}$ is the mass matrix of the system; $[C]_{17x17}$ is damping matrix; $[K]_{17x17}$ is a stiffness matrix. The displacement vector $\{q\}_{17x1}$ includes the displacements in the X-direction, Y-direction, as well as angular rotation, $\{F_i\}_{17x1}$ is excitation force vector:

$$\{F_{g_i}\}^T = [0\,0\,0\,0\,0\,0\,0\,m_1g\,m_2g\,m_3g\,m_4g\,m_5g\,m_6g\,m_7g\,\,0\,0\,0]$$

$$\{F_{u_i}\}^T = [0\,F_{u_{x2}}\,0\,F_{u_{x4}}\,0\,F_{u_{x6}}\,0\,0\,F_{u_{y2}}\,0\,F_{u_{y4}}\,0\,F_{u_{y6}}\,0\,F_{u_{\alpha2}}\,F_{u_{\alpha4}}\,F_{u_{\alpha6}}] \tag{10}$$

By solving equation (10) for i = 2, 4, 6; we obtain:

$$F_{u_{x_i}} = m_ie_i\,(\dot{\varphi}_i^2 \cos\varphi_i + \ddot{\varphi}_i \sin\varphi_i)$$

$$F_{u_{y_i}} = m_ie_i\left(\dot{\varphi}_i^2 \sin\varphi_i - \ddot{\varphi}_i \cos\varphi_i\right)$$

$$F_{u_{\alpha_i}} = m_ie_i\,(\ddot{x}_i \sin\varphi_i - \ddot{y}_i \cos\varphi_i)$$

### 2.3 Solution method

Using the Newmark time-stepping analysis method to solve Equation (9), the following variables and matrices are involved: $\{q_i\}$, $\{\dot{q}_i\}$, $[M]$, $[C]$, $[K]$, $\{F_g\}$, $\{F_u\}$, $\Delta t$, $t_i$, $\gamma$, $\beta$

Take the derivation of Equation (9), we have:

$$\{\ddot{q}_i\} = [M]^{-1}\left(-[C]\{\dot{q}_i\} - [K]\{q_i\} + \{F_g\} + \{F_u\}\right) \tag{11}$$

where

$$\{q_{i+1} = q_i + \Delta t\dot{q}_i + \left(\frac{1}{2} - \beta\right)\Delta t^2\ddot{q}_i + \beta\Delta t^2\ddot{q}_{i+1}\ \dot{q}_{i+1} = \dot{q}_i + (1-\gamma)\Delta t\ddot{q}_i + \gamma\Delta t\ddot{q}_{i+1}\ M\ddot{q}_{i+1} + C\dot{q}_{i+1} + Kq_{i+1} = F_{i+1} \tag{12}$$

Transforming Eq. (12), we obtain:

$$\left[M + C\gamma\Delta t + K\beta\Delta t^2\right]\ddot{q}_{i+1} = F_{i+1} - C\left[\dot{q}_i + (1-\gamma)\Delta t\ddot{q}_i\right] - K\left[q_i + \Delta t\dot{q}_i + + \left(\frac{1}{2} - \beta\right)\Delta t^2\ddot{q}_i\right] \tag{13}$$

Equation (13) can be generalized as:

$$\left[\widehat{K}\right]\{\ddot{q}_{i+1}\} = \left\{\widehat{F}_{i+1}\right\}$$

(14)

where $\left[\widehat{K}\right]$ is the effective stiffness matrix, determined as:

$$\left[\widehat{K}\right] = \left[M + C\gamma\Delta t + K\beta\Delta t^2\right]$$

(15)

$\left\{\widehat{F}_{i+1}\right\}$ is the effective force vector, determined as:

$$\widehat{F}_{i+1} = F_{i+1} - C\left[[\dot{q}_i + (1-\gamma)\Delta t\ddot{q}_i] - K\left[q_i + \Delta t\dot{q}_i + \left(\frac{1}{2} - \beta\right)\Delta t^2\ddot{q}_i\right]\right]$$

(16)

In this study, the implicit Newmark–β method was employed, since it does not require adherence to the time step condition as $\Delta t \le \frac{2}{\omega_{max}} = \frac{2}{319} = 0.006s$. The parameters $\gamma$ and $\beta$ were adjusted during the time-stepping loop and set to $\gamma = \frac{1}{2}$ and $\beta = \frac{1}{4}$. A constant time step of $\Delta t = 0.01s$ was selected for the simulations to ensure stability and computational efficiency [20]. Fig 5 shows the algorithm flowchart, used for computing the displacement $q_i$, the velocity $\dot{q}_i$, and the acceleration $\ddot{q}_i$.

## 3. Experimental procedure

An AISI 1045 steel transmission shaft as shown in Fig 3 with the unbalances $m_2$ and $m_6$ was prepared to investigate the behavior of nodes 1–7 during rotational operation. Table 1 shows the geometry dimensions and mechanical properties of the shaft [12,21,22].

Fig 6 shows the vibration testing machine, designed and fabricated for study the vibration behavior of the rotating shaft. The shaft was driven by an AC motor and an encoder was attached to the motor spindle to record the actual speed of the rotating shaft. A laser sensor measured the displacement of the rotor. The accelerometer and processor recorded the amount and positions of unbalance on the rotor. The operating tests were performed in controlled modes.

## 4. Results and discussions

### 4.1 The effect of rotational speed on shaft displacement

In Fig 7, the rotor oscillates when operated arround the first critical speed (mode 1). Under the effect of centrifugal force, the shaft is deflected and the shaft center will oscillate with displacement at position 4 with amplitude $y_{max}$.

Neglecting the damping coefficient C, the first critical speed of the system can be determined according to [23] using the following expression:

$$\omega_c = \sqrt{\frac{K_r}{m}} = \sqrt{\frac{7 * 10^7}{685}} = 319 \ rad/s$$

In this case, the critical speed is:

$$n_c = \frac{60 * \omega_1}{2 * \pi} = \frac{60 * 319}{2 * 3.14} = 3047 \ rpm$$

To ensure the stability of the rotor in the model and corresponding to the actual working speeds of the equipment, the experimental speed ranges of the rotor were selected as follows: $n_1 = 800$ rpm, $n_2 = 1500$ rpm and $n_3 = 2000$ rpm, in which $n_3$ is selected to be less than 70% of critical speed of 2132 rpm [24]. Fig 8 shows the displacement of node 4 in the X

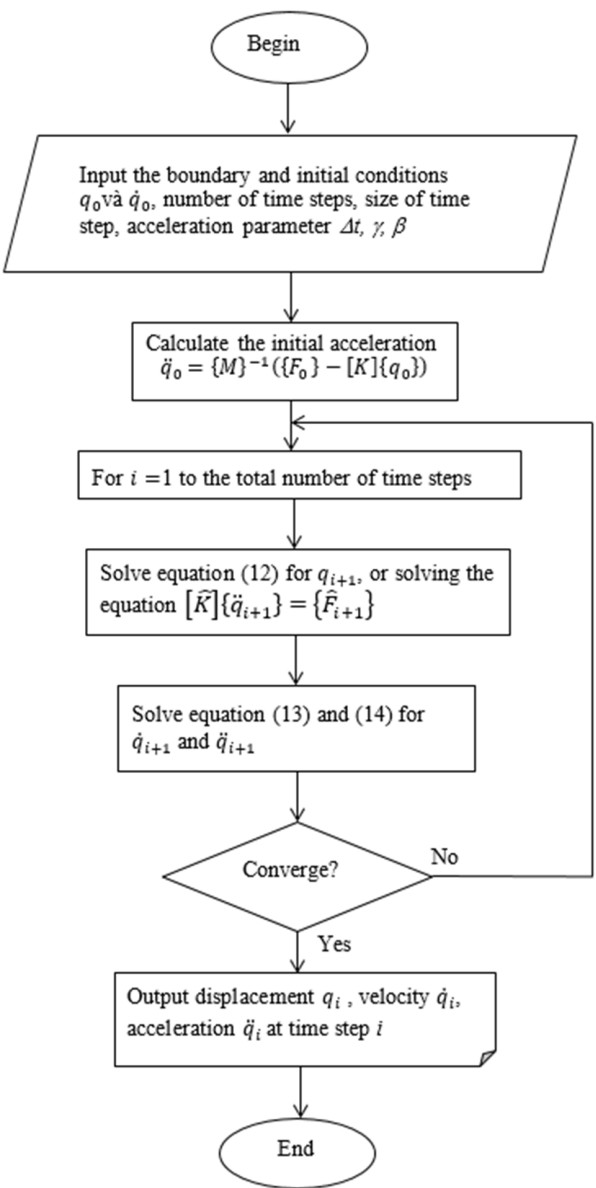

**Fig 5. A flowchart for dynamic response of the rotor system.**

direction at speed of $n_3 = 2000$ rpm, determined from the experiment and the Newmark–β numerical simulation. Generally, the amplitude measured from the experiment is larger than those using Newmark–β numerical computation because various factors in the manufacturing process may affect the unbalanced conditions.

Table 2 compares the X and Y displacement of node 4, determined from the Newmark-β method and from the experiment. The displacement determined experimentally is higher than those determined from the theoretical simulation because the actual stiffness of shaft and bearing is not uniform and due to the inaccuracy and clearance between shaft and bearing. For the eperation below the first critical speed, the vibrations remain stable, with displacement amplitudes of $X = \pm 0.046$ and $Y = \pm 0.081$. The relative error of displacement in the X- and Y-axes for speeds of 2000 rpm is 6% and 12%, respectively. As the rotor reaches the first critical speed of 3000 rpm, the vibrational amplitude significantly increases,

**Table 1. Symbols, parameters, and unit of power transmission shaft system.**

| No. | Parameters | Symbols | Unit | Value |
|---|---|---|---|---|
| 1 | Shaft length | $L$ | m | 0.256 |
| 2 | Section lengths | $l_1=l_8$ | m | 0.0125 |
| 3 | Section lengths | $l_2=l_7$ | m | 0.0405 |
| 4 | Section lengths | $l_3=l_6$ | m | 0.0325 |
| 5 | Section lengths | $l_4=l_5$ | m | 0.0425 |
| 6 | Shaft diameter | $d_{shaft}$ | m | 0.012 |
| 7 | Disk diameter | $D_{disk}$ | m | 0.1 |
| 8 | Disk thickness | $B$ | m | 0.02 |
| 9 | Initial phase angle of disk 1 | $\varnothing_{0_2}$ | rad | 0 |
| 10 | Initial phase angle of disk 2 | $\varnothing_{0_6}$ | rad | 0 |
| 11 | Unbalance eccentricity at node 2 | $e_2$ | m | $4.7 \times 10^{-3}$ |
| 12 | Unbalance eccentricity at node 6 | $e_6$ | m | $4.7 \times 10^{-3}$ |
| 13 | Stiffness of rotor | $K_r$ | N/m | $7 \times 10^7$ |
| 14 | Stiffness of bearing | $K_b$ | N/m | $7.2 \times 10^6$ |
| 15 | Stiffness of torsional shaft | $K_t$ | Nm/rad | $1 \times 10^5$ |
| 16 | Damping of bearing | $C$ | N.m/s | 20 |
| 17 | Friction coefficient | $\mu$ |  | 0.1 |
| 18 | Poisson's ratio | $\upsilon$ |  | 0.3 |
| 19 | Density | $\rho$ | kg/m³ | $7.8 \times 10^3$ |
| 20 | Shaft mass | $m_{shaft}$ | g | 285 |
| 21 | Disk mass | $m_{disk}$ | g | 200 |
| 22 | Rotor mass | $m$ | g | 685 |
| 23 | Masses of nodes 1 and 7 | $m_1=m_7$ | g | 15 |
| 24 | Masses of nodes 2 and 6 | $m_2=m_6$ | g | 35 |
| 25 | Masses of nodes 3 and 5 | $m_3=m_5$ | g | 80 |
| 26 | Mass of node 4 | $m_4$ | g | 25 |
| 27 | Gravitational acceleration | $g$ | m/s² | 9.81 |
| 28 | Moment of inertia | $I_p$ | m⁴ | $1.2 \times 10^{-7}$ |
| 29 | Young's modulus | $E$ | N/m² | $2.1 \times 10^{11}$ |
| 30 | Torsional modulus | $G$ | N/m² | $7.7 \times 10^{10}$ |
| 31 | Damping of torsional shaft | $C_t$ | Ns/rad | 0 |
| 32 | Angular velocity | $\omega_3$ | rad/s | 209 |

with displacements of X=±0.101 and Y=±0.119 mm because of the resonance effects of the rotor components. In general, the displacement in the direction Y is higher than in the X direction because of the gravity force of the components, acting downward, resulting in the higher vertical vibration. The relative errors in displacements are higher by 12% and 18%.

A comparison between the numerical results obtained using the Newmark–β method and the experimental data indicated that the displacement deviation of approximately 6%, which is acceptable for engineering analysis and validates the accuracy of the computational model.

## 4.2 The effect of unbalance on shaft displacement

In engineering, for some design circumstance or unavoidable eccentricity in the manufacturing process, the unbalance exists and should be taken into account. The unbalance, eccentricity, and angular position (the angle between the planes

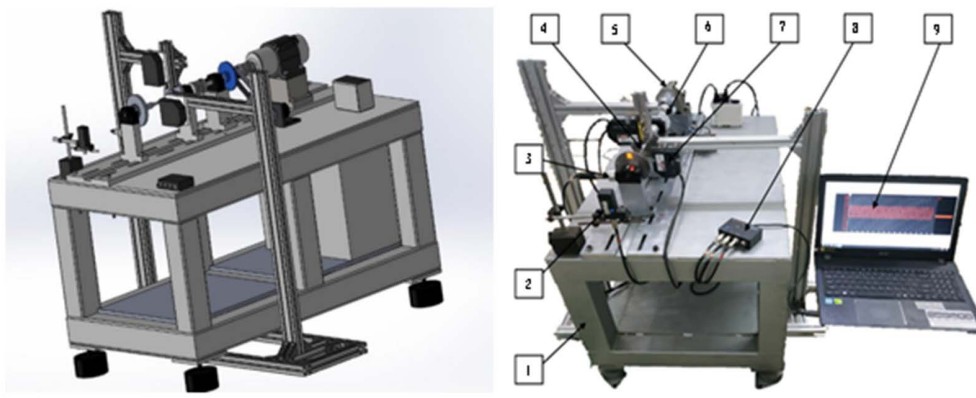

*a) Model 3D*  *b) Measurement setup of the rotating shaft*

1. Machine Pedestal – 2. Optical Sensor – 3. Acceleration Sensor – 4. Rotating shaft - 5. Motor – 6. Encoder –7. Measurement sensor LK-G35 – 8. DAQ – 9. Personal Computer

**Fig 6. A vibration testing machine of rotating shaft.**

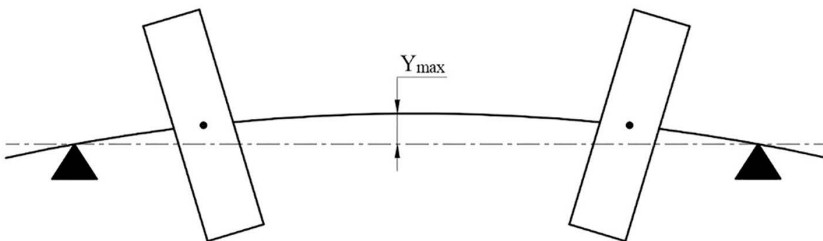

**Fig 7. A schematic plot of a rotor operating at the first critical speed (mode 1).**

of trial masses $m_2$ and $m_6$) will be analyzed hereafter. The masses $m_2$ and $m_6$ were mounted at three positions $0^0$–$180^0$, $0^0$–$90^0$, and $0^0$–$0^0$ such that the direction of $m_6$ respectively makes angular positions of $180^0$, $90^0$ and $0^0$ to the direction of $m_2$, as shown in Fig 9.

**4.2.1 Symmetrical trial mass (angular position: $0^0$–$0^0$ on both disks).** The structure of the unbalanced rotor is shown in Fig 9, where the trial masses $m_2$ and $m_6$ are in the same direction. According to the unbalance standard, the corresponding test masses considered in the range follows as G1, G2.5, G6.3 and G16 [25]. For G1, the allowable eccentricity e ($n_3 = 2000$ rpm) is:

$$e = \frac{G}{\omega} = \frac{1}{209} = 4.78 * 10^{-3} \; m$$

At the minimum speed $n_1$, the allowable test mass for G16 is calculated as follows:

$$m_{trial} = \frac{k * 9.54 * G_{16} * M_{rotor}}{n_1 * r} = 26.1 \; g$$

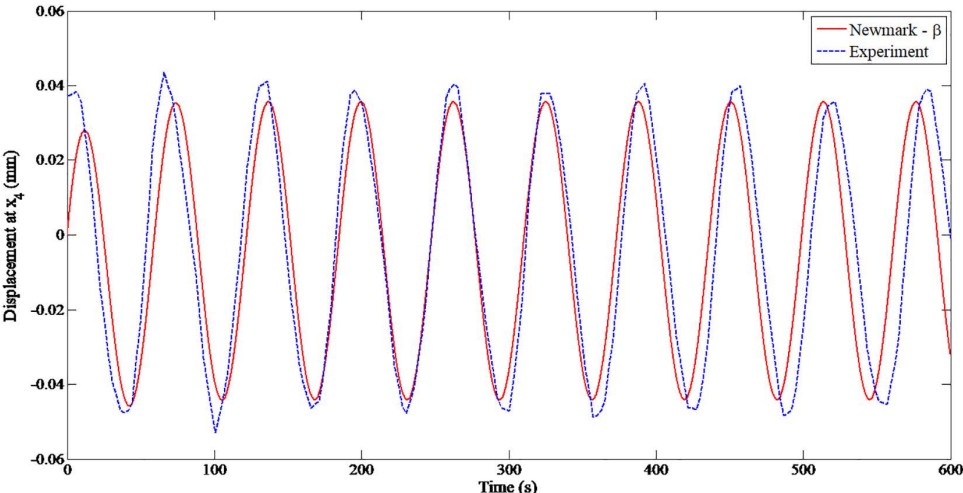

**Fig 8. Displacement of node 4 along X-axis at 2000 rpm.**

**Table 2. Displacement in the X and Y axes at node 4.**

| Frequency (Hz) | Speed (rpm) | Newmark-β method | | Experimental data | | Relative Error | |
|---|---|---|---|---|---|---|---|
| | | X (mm) | Y (mm) | X (mm) | Y (mm) | X | Y |
| 34 | 2000 | ± 0.043 | ± 0.071 | ± 0.046 | ± 0.081 | 6% | 12% |
| 51 | 3000 | ± 0.085 | ± 0.097 | ± 0.101 | ± 0.119 | 15% | 18% |

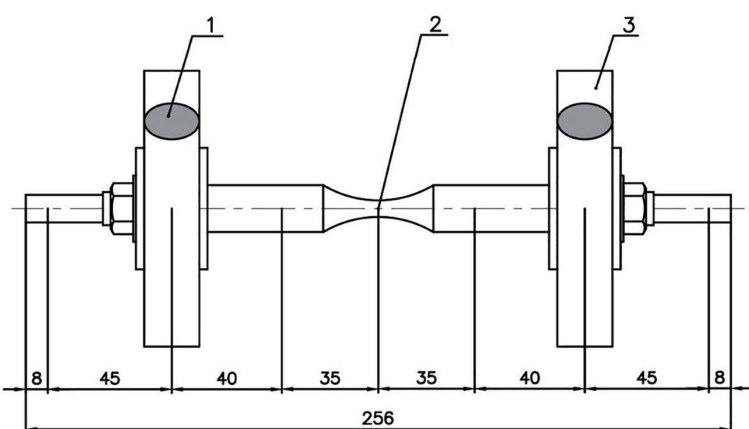

**Fig 9. Dimension of experimental unbalanced specimen.**

Three kinds of trial masses 10 g, 20 g, and 30 g and rotational speeds of 800, 1500 and 2000 rpm were selected to determine the unbalanced, using the commercial Erbessd – Instruments interface, as shown in Fig 10. For the highest trial mass $m_{trial}$ of 30 grams, the unbalanced amounts are respectively 41.18 mm/s and 39.06 mm/s for two nodes 2 and 6.

Table 3 represents the measurement results of rotor unbalance at angular position $0^0$–$0^0$ with corresponding to load values and speeds. The corresponding unbalance significantly increases with the increase of trial mass $m_{trial}$ and the

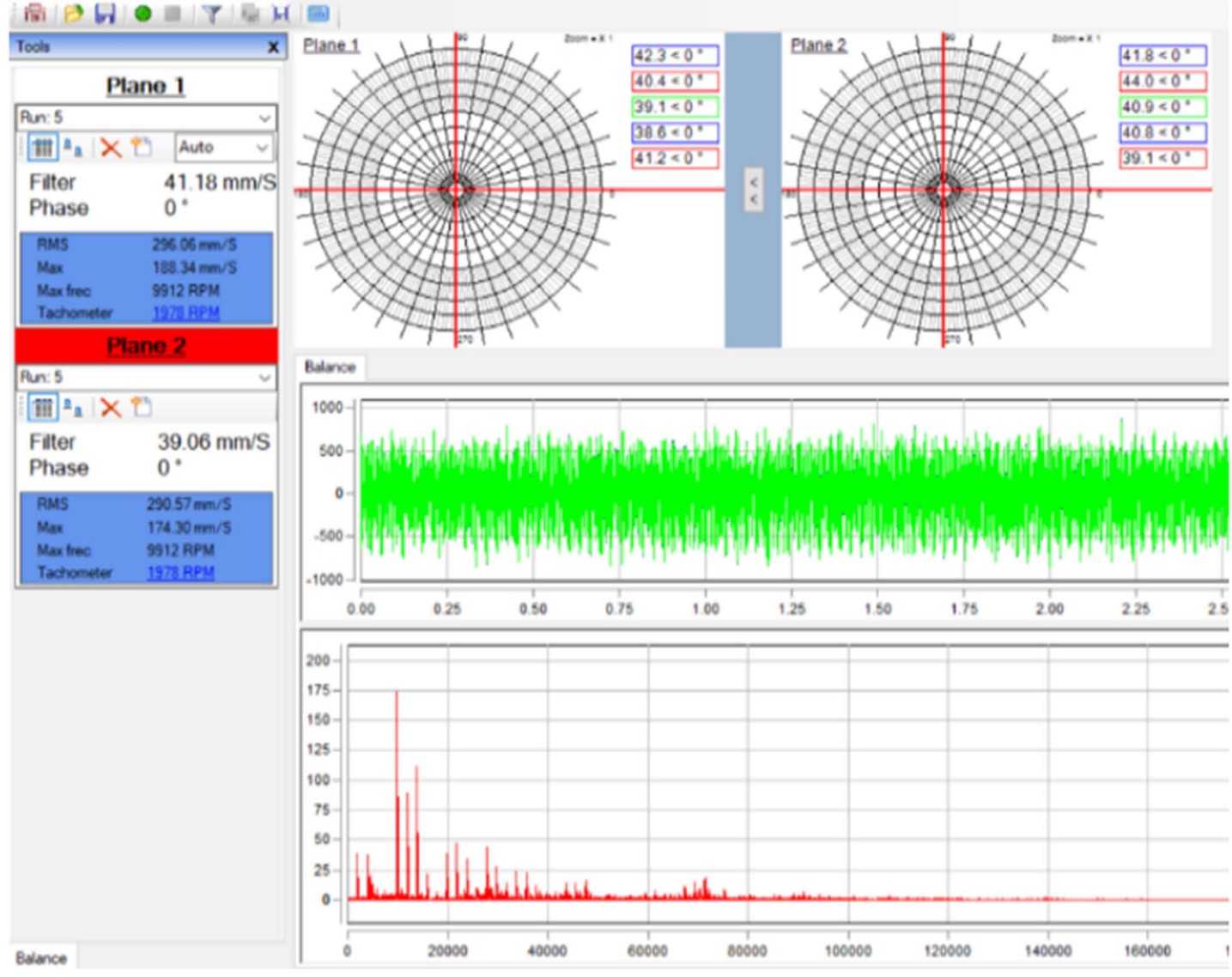

**Fig 10. The measurement results of balancing used to two planes at the angular position $0^0-0^0$; *mtrial* = 30 grams.**

rotational speed $n$ respectively. The unbalanced amount $G$ was highest of approximately 41.1 mm/s at $n$ of 2000 rpm with $m_{trial}$ of 30 g and got lowest of 0.58 mm/s at 800 rpm and $m_{trial}$ of 0 g.

Fig 11 and Table 4 represent the variation in horizontal X-axis displacement at node 4 under different operating speeds. The results clearly show the displacement amplitude increase with the rotational speed, particularly as the rotor approaches its first critical speed. This behavior reflects a corresponding rise in dynamic excitation, which is characteristic of resonance phenomena in rotor-dynamic systems.

**4.2.2 Asymmetrically placed trial mass (angular positions $0^0-90^0$ and $0^0-180^0$).** By changing the initial phase angle, the results of rotor unbalance at positions $0^0-90^0$ and $0^0-180^0$ are shown in Table 5. At a constant rotational speed of 2000 rpm, variations in the load angular position $0^0-0^0$, $0^0-90^0$, and $0^0-180^0$ and trial mass $m_{trial}$ of 10, 20, and 30g result

Table 3. Unbalance measurement results when attaching trial mass at angular position 0⁰–0⁰.

| Trial mass $m_{trial}$ (grams) | Speed (rpm) | Amount of unbalance $G$ (mm/s) |
|---|---|---|
| 0 | 800 | 0.58 |
| | 1500 | 1.12 |
| | 2000 | 1.34 |
| 10 | 800 | 0.9 |
| | 1500 | 4.9 |
| | 2000 | 20.5 |
| 20 | 800 | 1.1 |
| | 1500 | 9.4 |
| | 2000 | 32.8 |
| 30 | 800 | 2.2 |
| | 1500 | 14.3 |
| | 2000 | 41.1 |

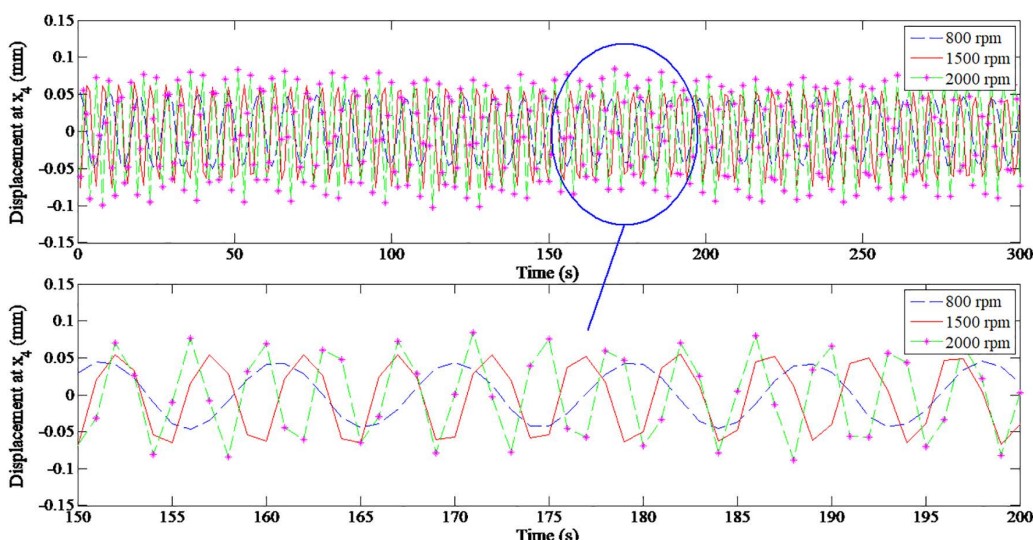

Fig 11. Horizontal displacement of node 4 for *mtrial* = 20 g at various speeds.

Table 4. Displacement measurement results at node 4 for 20g trial mass.

| $m_{trial}$ (grams) | Speed (rpm) | Displacement at $x_4$ (mm) |
|---|---|---|
| 20 | 800 | ± 0.055 |
| | 1500 | ± 0.073 |
| | 2000 | ± 0.098 |

**Table 5. Unbalance $G$ and displacement at 2000 rpm for various trial masses and angular positions.**

| Angular Position | Trial mass $m_{trial}$ (grams) | Amount of unbalance $G$ (mm/s) | Displacement at $x_4$ (mm) | Displacement at $y_4$ (mm) |
|---|---|---|---|---|
| Original (without trial mass) | 0 | 1.34 | ± 0.046 | ± 0.081 |
| $0^0 – 0^0$ | 10 | 20.5 | ± 0.073 | ± 0.100 |
| $0^0 – 90^0$ | | 14.3 | ± 0.064 | ± 0.098 |
| $0^0 – 180^0$ | | 2.6 | ± 0.050 | ± 0.085 |
| $0^0 – 0^0$ | 20 | 32.8 | ± 0.098 | ± 0.111 |
| $0^0 – 90^0$ | | 28.7 | ± 0.079 | ± 0.105 |
| $0^0 – 180^0$ | | 3.7 | ± 0.054 | ± 0.090 |
| $0^0 – 0^0$ | 30 | 42.3 | ± 0.109 | ± 0.135 |
| $0^0 – 90^0$ | | 35.4 | ± 0.090 | ± 0.111 |
| $0^0 – 180^0$ | | 4.2 | ± 0.054 | ± 0.096 |

in corresponding changes in the component unbalance. The experimental results indicate that the lowest unbalance of 2.6 mm/s occurs at a trial mass of 10 grams and angular positioned at $0^0–180^0$. This is because the centrifugal forces of $m_2$ and $m_6$ canceled out. On the contrary, the unbalance $G$ obtained the highest value of 42.3 mm/s at the mass 30g and angular positioned at $0^0–0^0$ because of the resonance of centrifugal forces of $m_2$ and $m_6$.

In addition, Fig 12 and Table 6 show a decreasing trend of displacement amplitudes by ±0.098 mm, ±0.079 mm, and ± 0.054 mm as the load position shifts from $0^0–0^0$ to $0^0–90^0$ and then to $0^0–180^0$, respectively.

### 4.3 Motional orbit

To evaluate the influence of the unbalanced mass position on the rotor's motion trajectory, a series of experiments was conducted at a rotational speed of 2000 rpm. A trial mass 30g was mounted on the two discs of the rotating shaft, with various combinations of phase angular positions between the loads. Three unbalanced mass distribution cases were

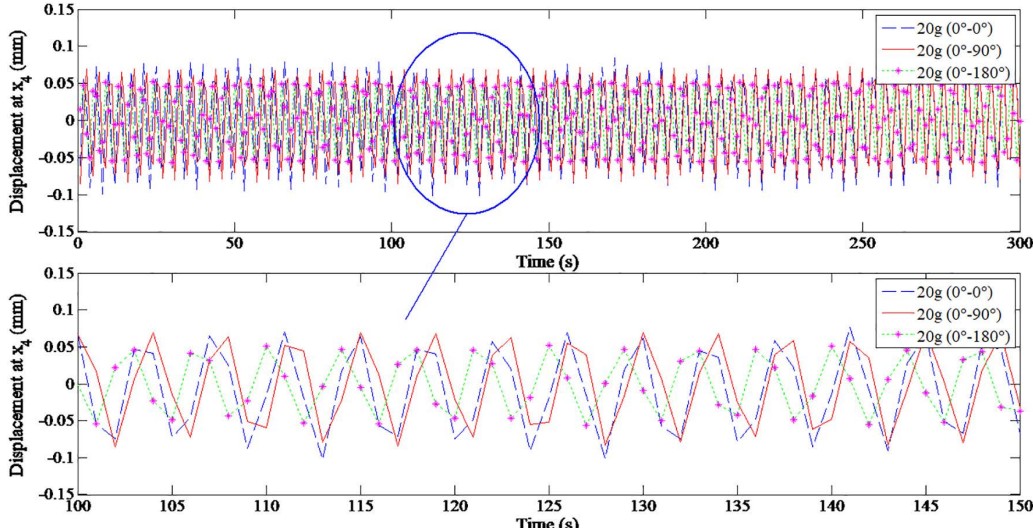

**Fig 12. Horizontal displacement of node 4 under 20 g trial mass and speed of 2000 rpm, for the different angular positions: $0^0–0^0$, $0^0–90^0$, and $0^0–180^0$.**

**Table 6. Displacement measurement results under 20g load at 2000 rpm, with different load angular positions: $0^0$–$0^0$, $0^0$–$90^0$, and $0^0$–$180^0$.**

| Speed (rpm) | Position | Displacement at $x_4$ (mm) |
|---|---|---|
| 2000 | $0^0 - 0^0$ | ± 0.098 |
| | $0^0 - 90^0$ | ± 0.079 |
| | $0^0 - 180^0$ | ± 0.054 |

investigated for three angular positions $0^0$–$0^0$, $0^0$–$90^0$ and $0^0$–$180^0$. The motion orbit of the shaft center at the speed $n_3 = 2000$ rpm is represented in Fig 13. The orbits for load-free rotation and the unbalanced masses in the opposite angular directions $0^0$–$180^0$ obtains the smallest value, while the orbit for angular position $0^0$–$0^0$ is most unstable. This is because of the cancelation or resonance of centrifugal forces of the masses, well agreeing with the experimental displacements in Table 5. Furthermore, the data indicate that the oscillation observed with a trial mass of 30g for the angular position $0^0$–$0^0$ closely resembles the shaft behavior at its first critical speed.

It can be observed that changes in the angular positions of the trial mass 0°–0°, 0°–90°, and 0°–180° strongly affect the vibration amplitude, thus the shape of the shaft center orbit and well agrees to the expected behavior predicted based on the influence parameters. The greater amount of unbalance and their positions concentration will generate stress accumulation and micro-cracks, resulting in shaft failure.

## 5. Conclusions

The following conclusions are made:

a. The effects of such key parameters as rotation speed, eccentricity due to unbalanced and initial phase angle during operation were simulated. The experimental equipment was built and experimental results were compared with the simulated data.

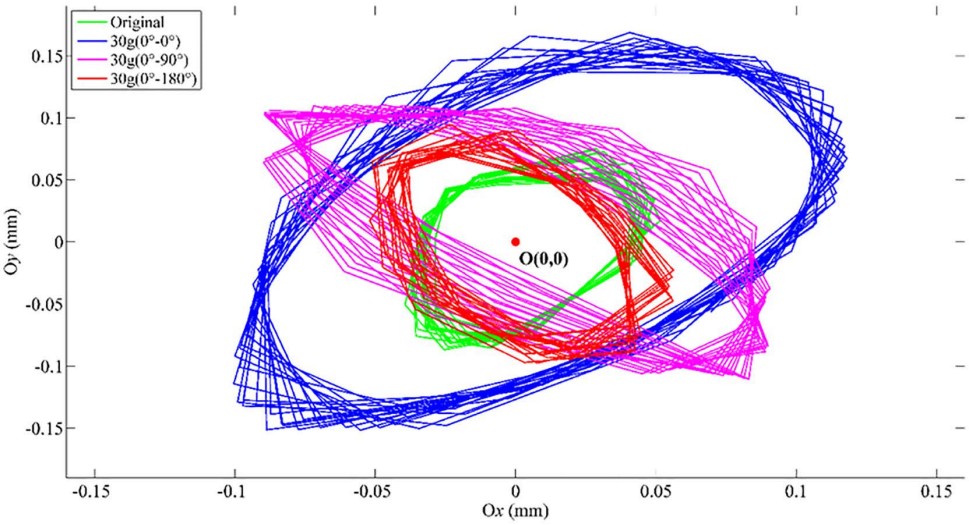

**Fig 13. Motional orbit of the shaft center at 2000 rpm for various angular positions.**

b. The vibration becomes severely unstable when rotation speed approaches the critical speed, as shown in Tables 3 and 4. The displacement amplitude significantly increases with the amount of unbalance, as shown in Table 5.

c. The initial phase angle of the trial mass position has great influence on the vibration of the rotating shaft, thus the fatigue bending strength and performance of the component, as shown in Tables 5 and 6 and Fig 13. By determining the vibration and unbalance conditions of the shaft component, the fatigue limitation can be predicted to ensure the safety and long-term stability of the rotor system.

d. For operation under the critical speed, the unbalance analysis using the Newmark-β method well agrees with the experimental result, as represented in Table 2, showing that the Newmark-β method is a reliable and stable approach for analyzing the behavior of rotor-bearing systems. This allows rapid and high computational efficiency to predict the shaft operation with fast convergence time.

Further research may be proceeded on the following issues:

i. Unbalance analysis for double-phase and composite materials.

ii. Analysis for materials with thin films, coating layer or the surface layer with residual stress.

iii. Unbalance analysis in high temperature conditions.

## Supporting information

**S1 Data. Experimental data for Figs 8, 11, 12 and 13.**
(XLSX)

## Acknowledgments

The authors would like to express their appreciation to the staff of the Metallurgy Laboratory at Ho Chi Minh City University of Technology and Education for their assistance.

## Author contributions

**Data curation:** Thanh Lam Tran, Vinh Phoi Nguyen.

**Formal analysis:** Thanh Lam Tran.

**Funding acquisition:** Thanh Lam Tran.

**Investigation:** Thanh Lam Tran.

**Methodology:** Thanh Lam Tran.

**Project administration:** Thanh Lam Tran.

**Writing – original draft:** Thanh Lam Tran, Chi Cuong Le.

**Writing – review & editing:** Thien Ngon Dang.

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
