## [Decision Letter · Decision Letter 0]

25 Apr 2025

Dear Dr. Dang,

Thank you for submitting your manuscript to PLOS ONE. After careful consideration, we feel that it has merit but does not fully meet PLOS ONE’s publication criteria as it currently stands. Therefore, we invite you to submit a revised version of the manuscript that addresses the points raised during the review process.

We look forward to receiving your revised manuscript.

Kind regards,

Ha Quang Thinh Ngo

Academic Editor

PLOS ONE

Journal Requirements:

4. Please note that funding information should not appear in any section or other areas of your manuscript. We will only publish funding information present in the Funding Statement section of the online submission form. Please remove any funding-related text from the manuscript.

6. We note that your Data Availability Statement is currently as follows: All relevant data are within the manuscript and in Supporting Information files.

7. Please amend the manuscript submission data (via Edit Submission) to include author Vinh Phoi Nguyen.

Additional Editor Comments:

Please carefully read the comments of reviewers. In the submission of revised manuscript, authors must have a response for each comment.

Reviewers' comments:

Reviewer's Responses to Questions

**Comments to the Author**

1. Is the manuscript technically sound, and do the data support the conclusions?

Reviewer #1: Partly

Reviewer #2: Yes

Reviewer #3: Yes

2. Has the statistical analysis been performed appropriately and rigorously?

Reviewer #1: Yes

Reviewer #2: Yes

Reviewer #3: Yes

3. Have the authors made all data underlying the findings in their manuscript fully available?

Reviewer #1: Yes

Reviewer #2: Yes

Reviewer #3: Yes

4. Is the manuscript presented in an intelligible fashion and written in standard English?

Reviewer #1: Yes

Reviewer #2: No

Reviewer #3: No

Reviewer #1: 1. Please explaing in detail why 3 and 6 set zero ( 3= 3=0)?. And in this paragraph: “Here, α3 = α6 = 0; at this node, the remaining 3 positions 2, 4, and 6 need to be investigated: = [ , , ] where j = 2, 4, 6”. Author set 6 = 0, but the �6 are investigted. Please explaining more this problem?

2. Retype the symbol of “Poisson coefficient” in Table 1.

3. In this studay, Did author used Implicit or Explicit Newmark-β method?

4. In the investigation of “The Effect of Unbalance on Shaft Displacement”, results of displacement on the X-axis and Y-axis were obtained from Experimental Equipment or from numerial analysis? Why don’t show the comparison of Simulation and Experiment results in this investigation.

5. This study conclued that “These research results demonstrate that Newmark - β method used to analyzing the behavior of rotor systems with the reliability and high stability, fast convergence time and accuracy to 5%.”. Where did this conclusion show in paper?

Reviewer #2: + Your English is not good. Authors should use the Professional English Editing Service to improve your writing. In the submission of revision, authors must submit the certificate of this service.

+ Keyword section contain at least five key words

+ Authors should use some potential tools to check AI in order to ensure the free-AI writing

+ In each figure, components must be tagged by numbers or characters. Then, in the body text, authors must explain these components in details.

+ Each mathematical symbols in figures must be described in table or text

+ Remember that in the multiple images of one figure, each one must be labelled by (a), (b) or (c). Later, in the caption of figure, the description of these images should be denoted

+ Characters in image are too small, it is difficult for readers to understand

+ Fig. 6 should be removed since there is no meaningful. Authors stated that they indicate the dimensions of transmission shaft, however only lengths of them are displayed

+ What is relation between Fig. 6 and Table 1?

+ It is hard to read the parameters in both x axis and y axis

+ In Fig. 12, what does it mean, how can readers understand

+ Please improve the quality of all figures in this manuscript

+ I think that authors did not prepare this manuscript carefully

Reviewer #3: + Abstract must be re-written so that it reflects your works with significant contributions, briefly describe your method and insists on what technique you did

+ Title is so boring, authors should use native English speaker to modify some errors in grammar and phrase

+ Reference section is very poor, commonly there are necessary to cite 30-40 publications in related fields. Hence you must invest more efforts to read additional papers. Some key developments such as 10.1177/1077546320953733, 10.1007/s11071-022-08100-3, 10.1016/j.ymssp.2023.110941 and 10.1016/j.ymssp.2022.109691 should be mentioned and analyzed in detail.

+ Cited style should be changed. Term "author et al" is repeated so many times and should be replaced by the others

+ In the end of Introduction section, the contributions of this work must be declared

+ Also, the structure of this manuscript should be mentioned shortly at the end of introduction

+ All equations must be tagged in the order

+ Fig. 1 is very blurred and authors must mention each component of this figure in the body text

+ Fig. 2 is very basic and there is no need to display in the scientific research

+ In one figure, if there are more than two images, authors must tag each individually and explain them in the caption of figure

+ All mathematical symbols must be explained in detail such as x1, x2 ....

+ Fig. 4 lacks of the interactive forces/moments so that our readers could analyze your system

+ Where is equation (8) derived ? which theory or theorem. Why vector of F force appear in the explanation but there is no vector in the computational equation

+ In equation (3), what are these matrices and vectors? there is no explanation or explicit representation

+ It looks like that authors did not understand clearly what theoretical development was developed in this work

+ Why Table 5 is only validated in one mode such as speed 2000 rpm

**Do you want your identity to be public for this peer review?** For information about this choice, including consent withdrawal, please see our Privacy Policy

Reviewer #1: No

Reviewer #2: No

Reviewer #3: No

---

## [Author Response · Author response to Decision Letter 1]

12 Jun 2025

Dear Editor and Reviewer,

The revised paper is submitted. The corrections made in the revised paper are colored in red. Much appreciate your comments and help in the preparation of the paper.

Please find our responses to the reviewers' comments in the attached file named 'Response to Reviewers

Kind regards,

Ngon Dang Thien

---

## [Decision Letter · Decision Letter 1]

11 Aug 2025

Dear Dr. Dang,

Thank you for submitting your manuscript to PLOS ONE. After careful consideration, we feel that it has merit but does not fully meet PLOS ONE’s publication criteria as it currently stands. Therefore, we invite you to submit a revised version of the manuscript that addresses the points raised during the review process.

We look forward to receiving your revised manuscript.

Kind regards,

Ha Quang Thinh Ngo

Academic Editor

PLOS ONE

Journal Requirements:

Reviewers' comments:

Reviewer's Responses to Questions

**Comments to the Author**

Reviewer #2: All comments have been addressed

2. Is the manuscript technically sound, and do the data support the conclusions?

Reviewer #2: Partly

3. Has the statistical analysis been performed appropriately and rigorously?

Reviewer #2: Yes

4. Have the authors made all data underlying the findings in their manuscript fully available?

Reviewer #2: Yes

5. Is the manuscript presented in an intelligible fashion and written in standard English?

Reviewer #2: No

Reviewer #2: + English is not good, authors should use the Professional English Editing Service to modify. In the submission of the revised manuscript, the certificate of this service should be submitted

+ Table 1 which indicate the system parameters of simulations, must be proved the correctness and properness. Authors must cite the reference source where you get them

+ Authors should produce a Table for abbreviation and symbol

+ Supporting information is very strange, authors must follow the instructions of this journal

+ The statement of Funding must contain the project code

+ Conclusion section must contain future works. Authors must specify the potential developments for our readers

+ Ref no 19, 21, 22 must be replaced by recent works

+ Results are presented, but there is limited discussion comparing them with similar studies in the literature.

**Do you want your identity to be public for this peer review?** For information about this choice, including consent withdrawal, please see our Privacy Policy

Reviewer #2: No

---

## [Author Response · Author response to Decision Letter 2]

16 Sep 2025

Dear Editor and Reviewers:

The revised paper is submitted. The corrections made in the revised paper are colored in red. Much appreciate your comments and help in the preparation of the paper.

List of revisions in response to the comments from reviewer #2

Reviewer #2:

1. English is not good, authors should use the Professional English Editing Service to modify. In the submission of the revised manuscript, the certificate of this service should be submitted

We sincerely appreciate the reviewer’s comment regarding the language quality. The authors have significantly revised the English text and explanation to the professional form.

2. Table 1 which indicate the system parameters of simulations, must be proved the correctness and properness. Authors must cite the reference source where you get them

We thank the reviewer for this important comment. Accordingly, the authors have added three references ([21], [22] and [23]) for the model parameters in the revised manuscript.

3. Authors should produce a Table for abbreviation and symbol

We would like to thank the reviewer for the valuable comment. In response, we have updated Table 1 in the revised manuscript to clarify the system parameters.

4. Supporting information is very strange, authors must follow the instructions of this journal

Thanks for comment, the supporting information has been re-formatted in the revised manuscript.

5. The statement of Funding must contain the project code

We highly appreciate the reviewer’s comment. The project code has been added to the funding statement in the revised manuscript.

6. Conclusion section must contain future works. Authors must specify the potential developments for our readers

We thank the reviewer for the helpful suggestion. We have included a brief discussion of potential future works in the conclusion section to highlight possible directions for further research.

7. Ref no 19, 21, 22 must be replaced by recent works

Thanks to the reviewer's comment, accordingly, references [19], [21] and [22] have been updated to [19], [24] and [25].

8. Results are presented, but there is limited discussion comparing them with similar studies in the literature.

We thank the reviewer for this valuable comment. In the revised manuscript, we have updated the Introduction section to provide a more comprehensive overview of previous studies and to emphasize that there are currently very few experimental works in this field. As a result, direct comparisons with existing studies are limited. Nevertheless, the experimental results presented in this work offer reliable data and can serve as a valuable reference for future research.

---

## [Decision Letter · Decision Letter 2]

15 Oct 2025

Investigation on the Influence of Unbalanced Shaft Component in Gearbox on Displacement Using the Newmark-β Method

PONE-D-25-01379R2

Dear Dr. Dang,

We’re pleased to inform you that your manuscript has been judged scientifically suitable for publication and will be formally accepted for publication once it meets all outstanding technical requirements.

Kind regards,

Ha Quang Thinh Ngo

Academic Editor

PLOS ONE

Additional Editor Comments (optional):

According to the comments of reviewer, this manuscript can be accepted in the present form

Reviewers' comments:

Reviewer's Responses to Questions

**Comments to the Author**

Reviewer #2: All comments have been addressed

2. Is the manuscript technically sound, and do the data support the conclusions?

Reviewer #2: Yes

3. Has the statistical analysis been performed appropriately and rigorously?

Reviewer #2: Yes

4. Have the authors made all data underlying the findings in their manuscript fully available?

Reviewer #2: Yes

5. Is the manuscript presented in an intelligible fashion and written in standard English?

Reviewer #2: Yes

Reviewer #2: In the revised manuscript, all questions have been addressed. It can be published in the present form

**Do you want your identity to be public for this peer review?** For information about this choice, including consent withdrawal, please see our Privacy Policy

Reviewer #2: No

---

## [Editor Report · Acceptance letter]

PONE-D-25-01379R2

PLOS ONE

Dear Dr. Dang,

I'm pleased to inform you that your manuscript has been deemed suitable for publication in PLOS ONE. Congratulations! Your manuscript is now being handed over to our production team.

Kind regards,

on behalf of

Dr. Ha Quang Thinh Ngo

Academic Editor

PLOS ONE